# Isotopic Constraints on the Atmospheric Sources and Formation of Nitrogenous Species in Biomass-Burning-Influenced Clouds

Yunhua Chang[1, 2], Yanlin Zhang[1*], Jiarong Li[2], Chongguo Tian[3], Linlin Song[4], Xiaoyao Zhai[1], Wenqi Zhang[1], Tong Huang[1], Yu-Chi Lin[1], Chao Zhu[2], Yunting Fang[4], Moritz F. Lehmann[5], and Jianmin Chen[2*]

[1]Yale-NUIST Center on Atmospheric Environment, Nanjing University of Information Science & Technology, Nanjing 210044, China.

[2]Shanghai Key Laboratory of Atmospheric Particle Pollution and Prevention (LAP[3]), Department of Environmental Science & Engineering, Institute of Atmospheric Sciences, Fudan University, Shanghai 200433, China.

[3]Key Laboratory of Coastal Environmental Processes and Ecological Remediation, Yantai Institute of Coastal Zone Research, Chinese Academy of Sciences, Yantai 264003, China.

[4]CAS Key Laboratory of Forest Ecology and Management, Institute of Applied Ecology, Chinese Academy of Sciences, Shenyang 110016, China.

[5]Aquatic and Isotope Biogeochemistry, Department of Environmental Sciences, University of Basel, Basel 4056, Switzerland.

[*]Corresponding authors: Yanlin Zhang (dryanlinzhang@outlook.com) and Jianmin Chen (jmchen@fudan.edu.cn).

**Abstract**

Predicting tropospheric cloud formation and subsequent nutrient deposition relies on understanding the sources and processes affecting aerosol constituents of the atmosphere that are preserved in cloudwater. However, this challenge is difficult to address quantitatively based on the sole use of bulk chemical properties. Nitrogenous aerosols, mainly ammonium ($NH_4^+$) and nitrate ($NO_3^-$), play a particularly important role in tropospheric cloud formation. While dry and wet (mainly rainfall) deposition of $NH_4^+$ and $NO_3^-$ are regularly assessed, cloudwater deposition is often underappreciated. Here we collected cloudwater samples at the summit of Mt. Tai (1545 m above sea level) in Eastern China during a long-lasting biomass burning (BB) event, and simultaneously measured for the first time the isotopic compositions (mean $\pm$ 1σ) of cloudwater nitrogen species ($\delta^{15}$N-NH$_4^+$ = -6.53 $\pm$ 4.96‰, $\delta^{15}$N-NO$_3^-$ = -2.35 $\pm$ 2.00‰, $\delta^{18}$O-NO$_3^-$ = 57.80 $\pm$ 4.23‰), allowing insights into their sources and potential transformation mechanism within the clouds. Large contributions of BB to the cloudwater $NH_4^+$ (32.9 $\pm$ 4.6%) and $NO_3^-$ (28.2 $\pm$ 2.7%) inventories were confirmed through a Bayesian isotopic mixing model, coupled with our newly-developed computational quantum chemistry module. Despite an overall reduction in total anthropogenic $NO_x$ emission due to effective emission control actions and stricter emission standards for vehicles, the observed cloud $\delta^{15}$N-NO$_3^-$ values suggest that $NO_x$ emissions from transportation may have exceeded emissions from coal combustion. $\delta^{18}$O-NO$_3^-$ values imply that the reaction of OH with $NO_2$ is the dominant pathway of $NO_3^-$ formation (57 $\pm$ 11%), yet the contribution of heterogeneous hydrolysis of dinitrogen pentoxide was almost as important (43 $\pm$ 11%). Although the limited sample set used here results in a relatively large uncertainty with regards to the origin of cloud-associated nitrogen deposition, the high concentrations of inorganic nitrogen imply that clouds represent an important source of nitrogen, especially for nitrogen-

limited ecosystems in remote areas. Further simultaneous and long-term sampling of aerosol, rainfall, and cloudwater is vital for understanding the anthropogenic influence on nitrogen deposition in the study region.

## 1 Introduction

50   Nitrogenous aerosols, mainly nitrate ($NO_3^-$) and ammonium ($NH_4^+$), formed from the emissions of nitrogen oxides ($NO_x = NO + NO_2$) and ammonia ($NH_3$), are major chemical components of aerosols, which serve as cloud condensation nuclei (CCN) and thus play an important role during cloud formation in the troposphere (Gioda et al., 2011; van Pinxteren et al., 2016). Cloudwater-containing nitrogenous compounds also represent a vital contributor to
55 nitrogen (N) budgets of terrestrial (Li et al., 2016b; Liu et al., 2013; Weathers and Likens, 1997; Vega et al., 2019) and marine ecosystems (Kim et al., 2014; Okin et al., 2011). However, the sources and formation processes of cloudwater N species are only poorly understood.

   $NO_x$ can be emitted from both anthropogenic and natural sources. Globally, over 50% of the $NO_x$ emissions derive from combustion of fossil fuel ($\sim$ 25 Tg N yr$^{-1}$; Jaegle et al., 2005;
60 Richter et al., 2005; Duncan et al., 2016), with the remainder being primarily soil-related emissions ($\sim$ 9 Tg N yr$^{-1}$;Lamsal et al., 2011; Price et al., 1997; Yienger et al., 1995; Miyazaki et al., 2017), or deriving from biomass burning ($\sim$ 6 Tg N yr$^{-1}$), and lightning (2-6 Tg N yr$^{-1}$) (Anenberg et al., 2017; Levy et al., 1996). The atmospheric sinks of $NO_x$ include the production of $HNO_{3(g)}$ and the formation of aerosol $NO_3^-$ (Seinfeld and Pandis, 2012), the partitioning of which can vary with
65 time (Morino et al., 2006). As for $NH_3$, over 90% of the $NH_3$ emissions in terrestrial ecosystems originate from agricultural production, such as livestock breeding and $NH_3$-based fertilizer application (Paulot et al., 2014; Kang et al., 2016; Reis et al., 2009; Bouwman et al., 1997; Heald et al., 2012; Zhang et al., 2018; Balasubramanian et al., 2015; Huang et al., 2011). In the urban

atmosphere, recent studies suggest that non-agricultural activities like wastewater discharge

(Zhang et al., 2017), coal burning (Li et al., 2016a), solid waste (Reche et al., 2012), on-road traffic (Suarez-Bertoa et al., 2014), and green space (Teng et al., 2017) also contribute to $NH_3$ emissions. In reactions with $H_2SO_4$ and $HNO_3$, $NH_3$ contributes to the formation of $NH_4^+$ salts, which typically make up from 20 to 80% of fine particle ($PM_{2.5}$) in the atmosphere (Seinfeld and Pandis, 2012).

Biomass burning (BB) is an important source of N in the atmosphere (Lobert et al., 1990; Souri et al., 2017). During the harvest/hot season of eastern China, agricultural BB frequently occurs and modifies the concentration and composition of aerosols in the atmosphere (Chen et al., 2017; Zhang and Cao, 2015). For example, about 50% of the N derived from biomass combustion can be released as $NH_3$ and $NO_x$ to form particulate $NH_4^+$ and $NO_3^-$, which then account for over

80% of total nitrogenous species in BB smoke particles (Crutzen and Andreae, 1990). BB-induced aerosols have not only been associated with poor air quality and the detrimental effects on human health, they have also shown to exert manifold effects on tropospheric clouds, altering regional or even global radiation budgets (Chen et al., 2014; Norris et al., 2016; Voigt and Shaw, 2015).

       The optical and chemical properties of clouds (and thus their radiative forcing) are directly

related to the aerosol and precipitation chemistry (Seinfeld et al., 2016). Moreover, clouds represent reactors of multiphase chemistry, contributing to many chemical transformations that would otherwise not take place, or would proceed at much slower rates (Herrmann et al., 2015; Lance et al., 2017; Ravishankara, 1997; Schurman et al., 2018; Slade et al., 2017). Understanding the sources and fate of nitrogenous species in BB-influenced clouds is particularly important to

comprehensively assess the environmental impacts of BB. But this challenge is difficult to address based on the sole use of bulk chemical properties (as most often done in previous studies).

Given that the [15]N can be preserved between the sources and sinks of $NO_x$ and $NH_3$, the N isotopic composition of $NO_3^-$ ($\delta^{15}$N-$NO_3^-$) and $NH_4^+$ ($\delta^{15}$N-$NH_4^+$) can be related to different sources of $NO_x$ and $NH_3$, and thus delivers useful information regarding the partitioning of the origins of atmospheric/cloudwater $NO_x$ and $NH_3$, respectively (Hastings et al., 2013; Michalski et al., 2005; Morin et al., 2008; Chang et al., 2018). This is different for the O isotopes. $NO_3^-$ production involves the oxidation of NO. The first step in the overall process is the conversion of NO into $NO_2$, e.g., through the oxidation by either ozone ($O_3$) or peroxy radicals (Michalski et al., 2011). Significant [18]O enrichments and excess [17]O (i.e., clear evidence for mass independent fractionation) are observed in atmospheric $NO_3^-$ collected across the globe (e.g., Michalski et al., 2005; Hastings et al., 2003). Such diagnostic isotope signatures, as well as their variability in space and time have been linked to the extent of $O_3$ oxidation (Michalski et al., 2011). Put another way, the oxygen isotope composition of $NO_3^-$ ($\delta^{18}$O-$NO_3^-$) is largely determined by chemical reactions rather than the source, and it it is primarily modulated by the O-atom exchange (Michalski et al., 2011).in the atmosphere Therefore, $\delta^{18}$O-$NO_3^-$ has the potential to indicate the relative importance of various $NO_3^-$ formation pathways (i.e., oxidation pathways during conversion of nitrogen oxides to $NO_3$ ) (Alexander et al., 2009; Elliott et al., 2009).

The O isotope fractionation during the conversion of $NO_x$ to $HNO_3$/$NO_3^-$ ($\varepsilon_{O\left(NO_x\leftrightarrow HNO_3\right)}\Big/\varepsilon_{O\left(NO_x\leftrightarrow NO_3^-\right)}$) involves two oxidation pathways (Hastings et al., 2003)

$$
\begin{aligned}
\varepsilon_{O\left(NO_x\leftrightarrow NO_3^-\right)} &= \varepsilon_{O\left(NO_x\leftrightarrow HNO_3\right)} = \gamma \times \varepsilon_{O\left(NO_x\leftrightarrow NO_3^-\right)_{OH}} + \left(1-\gamma\right) \times \varepsilon_{O\left(NO_x\leftrightarrow pNO_3^-\right)_{H_2O}} \\
&= \gamma \times \varepsilon_{O\left(NO_x\leftrightarrow HNO_3\right)_{OH}} + \left(1-\gamma\right) \times \varepsilon_{O\left(NO_x\leftrightarrow HNO_3\right)_{H_2O}}
\end{aligned}
\tag{1}
$$

where $\gamma/(1-\gamma)$ represents the contribution ratio of the isotope fractionation associated with the formation of $HNO_3$/$NO_3^-$ through the "OH+$NO_2$" pathway ($\varepsilon_{O\left(NO_x\leftrightarrow NO_3^-\right)_{OH}}$) and the hydrolysis

of dinitrogen pentoxide ($N_2O_5$) ($\varepsilon_{O\left(NO_x \leftrightarrow NO_3^-\right)_{H_2O}}$), respectively. The $\delta^{18}O$ value of $HNO_3$ produced

by the former process reflects the O atom partitioning of 2/3 $O_3$ and 1/3 OH:

$$
\begin{aligned}
\varepsilon_{O\left(NO_x \leftrightarrow NO_3^-\right)_{OH}} &= \varepsilon_{O\left(NO_x \leftrightarrow HNO_3\right)_{OH}} = \frac{2}{3}\varepsilon_{O\left(NO_2 \leftrightarrow HNO_3\right)_{OH}} + \frac{1}{3}\varepsilon_{O\left(NO \leftrightarrow HNO_3\right)_{OH}} \\
&= \frac{2}{3}\left[\frac{1000\left(^{18}\alpha_{NO_2/NO} - 1\right)\left(1 - f_{NO_2}\right)}{\left(1 - f_{NO_2}\right) + \left(^{18}\alpha_{NO_2/NO} \times f_{NO_2}\right)} + \left(\delta^{18}O\text{-}NO_x\right)\right] + \\
&\quad \frac{1}{3}\left[\left(\delta^{18}O\text{-}H_2O\right) + 1000\left(^{18}\alpha_{OH/H_2O} - 1\right)\right]
\end{aligned}
\tag{2}
$$

As for the $\delta^{18}O$ value of $HNO_3$ formed during hydrolysis of $N_2O_5$, 5/6 of the O atoms is

derived from $O_3$ and 1/6 from OH (Hastings et al., 2003):

$$
\varepsilon_{O\left(NO_x \leftrightarrow NO_3^-\right)_{H_2O}} = \varepsilon_{O\left(NO_x \leftrightarrow HNO_3\right)_{H_2O}} = \frac{5}{6}\left(\delta^{18}O\text{-}N_2O_5\right) + \frac{1}{6}\left(\delta^{18}O\text{-}H_2O\right)
\tag{3}
$$

where $f_{NO_2}$ refers to the fraction of $NO_2$ in the total $NO_x$ pool. Values for $f_{NO_2}$ vary

between 0.2 and 0.95 (Walters and Michalski, 2015). $\delta^{18}O$-X is the O isotopic composition of X.

The range of $\delta^{18}O$-$H_2O$ can be approximated using an estimated tropospheric water vapor $\delta^{18}O$

range of -25‰-0‰ (Zong et al., 2017). The $\delta^{18}O$ of $NO_2$ and $N_2O_5$ varies between 90‰ and 122‰

(Zong et al., 2017). $^{18}\alpha_{NO_2/NO}$ and $^{18}\alpha_{OH/H_2O}$ represent the equilibrium O isotope fractionation

factor between $NO_2$ and NO, and OH and $H_2O$, respectively, which is temperature-dependent

$$
1000\left(^m\alpha_{X/Y} - 1\right) = \frac{A}{T^4} \times 10^{10} + \frac{B}{T^3} \times 10^8 + \frac{C}{T^2} \times 10^6 + \frac{D}{T} \times 10^4
\tag{4}
$$

where A, B, C, and D are experimental constants over the temperature range of 150-450 K. Based

on Equations 1-4 and measured values for $\delta^{18}O$-$NO_3^-$ of cloudwater, a Monte Carlo simulation was

performed to generate 10000 feasible solutions. The error between predicted and measured $\delta^{18}O$ was less than 0.5‰.

At the end of July 2015, a large-scale BB event occurred over eastern and northern China. We took advantage of this special event to collect cloudwater samples at a high-altitude mountaintop site in the North China Plain, and to calibrate the isotopic signatures that BB events leave in the N pool of clouds. Integrating cloudwater nitrogenous species isotope data ($\delta^{15}N\text{-}NH_4^+$, $\delta^{15}N\text{-}NO_3^-$ and $\delta^{18}O\text{-}NO_3^-$) in a Bayesian isotopic mixing model coupled with a newly developed

computational quantum chemistry module (Chang et al., 2018), and using an isotopic mass balance approach, the sources and production pathways of inorganic nitrogen in cloudwater were quantified. Although numerous studies have been conducted that involved the chemical characterization of fog water or cloudwater, to our knowledge, there are no reports on the N (and O) isotopic composition of both $NO_3^-$ *and* $NH_4^+$ in cloudwater.

**2 Materials and Methods**

**2.1 Cloudwater Sample Collection**

Mt. Tai (117°13′ E, 36°18′ N; 1545 m above sea level) is a world-recognized geopark of key natural, historical and cultural significance, located in the eastern North China Plain (Fig. 1). It belongs to China's most important agricultural and industrial production areas, and the

composition of the atmosphere near the mountain can be considered representative with regards to the quality and levels of atmospheric pollution in the region (Li et al., 2017; Liu et al., 2018). Given the opportunistic nature of this study, cloudwater sampling commenced at the summit of Mt. Tai three days after the fire began (08/01/2015 19:12 to 08/03/2015 6:12; Table 1). In total, six cloudwater samples were collected during the long-lasting cloud event using a single-stage

Caltech active strand cloud-water collector (CASCC), as described by (Demoz et al., 1996). The

cloud collector was cleaned prior to each sampling using high-purity deionized water. After

sampling, cloud samples were filtered immediately using disposable syringe filters (0.45 µm) to

remove any suspended particulate matter, and then stored in a freezer at -80 °C until further

analysis. More details on the monitoring site and sampling procedures can be found elsewhere (Li

et al., 2017).

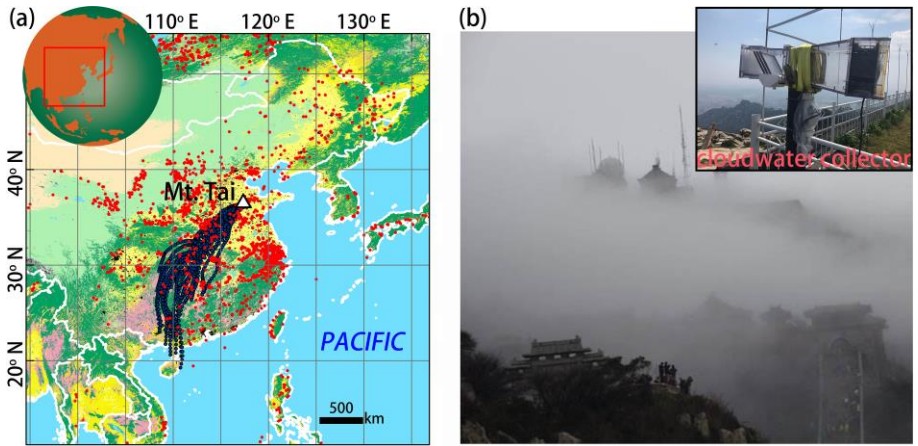

**Figure 1**. (**a**) Location of Mt. Tai (triangle) and twenty-four 48-h back trajectories (black

lines) of air masses simulated by the HYSPLIT model (http://ready.arl.noaa.gov/HYSPLIT.php)

based on the Global Data Assimilation System (GDAS) meteorological data set arriving at Mt. Tai

on 1 August 2015 (4:00 UTC) at an altitude 1500-m a.s.l (close to the altitude of our sampling

site). The base map of land use in China was modified from Chang et al. (2018). The red dots

represent the positions of wildfires between 29 and 31 July 2015, based on moderate-resolution

imaging spectroradiometer (MODIS) (http://modis-fire.umd.edu). (**b**) Field photos of clouds-

shrouded Mt. Tai and the cloudwater collector.

**2.2 Chemical and Isotopic Analysis**

Inorganic ions (including $SO_4^{2-}$, $NO_3^-$, $Cl^-$, $NH_4^+$, $K^+$, $Ca^{2+}$, $Mg^{2+}$, and $Na^+$), as well as levoglucosan, a specific tracer of biomass burning, in cloudwater samples were analyzed using a Dionex$^{TM}$ ICS-5000$^+$ system (ThermoFisher Scientific, Sunnyvale, USA). The IC system was equipped with an automated sampler (AS-DV). Cloudwater samples were measured using an

IonPac CG12A guard column and a CS12A separation column, with an aqueous methanesulfonic acid (MSA, 30 mM L$^{-1}$) eluent at a flow rate of 1 mL min$^{-1}$. Detailed information regarding sample processing, pre-treatment, chemical analyses, and analytical protocol adaption can be found elsewhere (Cao et al., 2016, 2017). The detection limits for $Na^+$, $NH_4^+$, $K^+$, $Mg^{2+}$, $Ca^{2+}$, $Cl^-$, $NO_2^-$, $NO_3^-$, $SO_4^{2-}$, and levoglucosan are 0.06, 0.03, 0.12, 0.08, 0.13, 0.64, 1.11, 2.67, 1.41, and 1.29 ppb

(part per billion by weight in solution), respectively. The analytical errors from duplicate analysis were within 5%.

Analysis of the isotopic compositions of $NH_4^+$ ($\delta^{15}N$-$NH_4^+$) and $NO_3^-$, ($\delta^{15}N$- $NO_3^-$ and $\delta^{18}O$-$NO_3^-$) was based on the isotopic analysis of nitrous oxides ($N_2O$) after chemical conversion of the respective target compound. More precisely, dissolved $NH_4^+$ in cloudwater samples was

oxidized to $NO_2^-$ by alkaline hypobromite ($BrO^-$), and then reduced to $N_2O$ by hydroxylamine hydrochloride ($NH_2OH.HCl$) (Liu et al., 2014). $NO_3^-$ was initially transformed to $NO_2^-$ by cadmium, and then further reduced to $N_2O$ by sodium azide ($NaN_3$) in an acetic acid buffer (McIlvin and Altabet, 2005; Tu et al., 2016). The produced $N_2O$ was analyzed using a purge and cryogenic trap system (Gilson GX-271, IsoPrime Ltd., Cheadle Hulme, UK), coupled to an isotope

ratio mass spectrometer (PT-IRMS) (IsoPrime 100, IsoPrime Ltd., Cheadle Hulme, UK). In order to correct for any machine drift and procedural blank contribution, international $NH_4^+$ (IAEA N1, USGS 25, and USGS 26) and $NO_3^-$ (IAEA N3, USGS 32, and USGS 34) standards were processed in the same way as samples. Standard regressions were made based on the known isotopic values

of international standards and the measured standard $\delta^{15}$N values. The slope of the plot of the sample versus the standard $\delta^{15}$N (0.49) was very close to the expected slope (0.5), which can be predicted based on the fact that half of the N atoms were derived from the azide (McIlvin and Altabet, 2005). The $r^2$ of the regression line was 0.999. The analytical precision for both multiple N and O isotopic analyses was better than 0.3‰ ($n = 5$).

## 2.3 Bayesian Mixing Model Analysis

By taking the uncertainty associated with the N isotopic signatures of multiple sources and associated isotope fractionation during (trans-)formations into account, the Bayesian method is more appropriate than simple linear mixing modeling to yield estimates on the source partitioning of a mixture like air pollutants (Chang et al., 2016; Chang et al., 2018). The relative contribution of each source in Bayesian theorem is expressed as:

$$P\left(f_q \big| data\right) \; = \; \theta\left(data \big| f_q\right) \times P\left(f_q\right) \Big/ \sum \theta\left(data \big| f_q\right) \times P\left(f_q\right) \quad (5)$$

where $\theta(data|f_q)$ and $P(f_q)$ represent the likelihood of the given mixed isotope signature, and the pre-determined probability of the given state of nature, based on prior information, respectively. The denominator represents the numerical approximation of the marginal probability of the data. Here the Bayesian mixing model MixSIR (stable isotope mixing models using sampling-importance-resampling) was used to disentangle the various potential NH$_3$ and NO$_x$ sources contributing to the cloudwater NH$_4^+$ and NO$_3^-$ pools, respectively, by forming the true probability distributions through generating 10000 solutions of source apportionment. Details on the model approach can be found in Appendix A.

The measured $\delta^{15}$N-NO$_3^-$ values of cloudwater samples depend on the $\delta^{15}$N signatures of the original NO$_x$ sources ($\delta^{15}$N-NO$_x$), the N isotope fractionation between nitrogen oxides (i.e.,

NO and NO$_2$ (Walters et al., 2016)), and the N isotope enrichment factor ($\varepsilon_N$) associated with the kinetic transformation of NO$_x$ to HNO$_3$ (Walters and Michalski, 2015). $\varepsilon_N$ is considered a hybrid of two dominant processes: one is the reaction of NO$_2$ and OH radicals to form NO$_3^-$, the other is the heterogeneous hydrolysis of dinitrogen pentoxide (N$_2$O$_5$) with water to form NO$_3^-$. We recently

developed a quantum chemistry computation module to quantify the N fractionation during nitrate formation, which had been validated by field measurements (Chang et al., 2018). Here this module was adopted to calculate the N isotope fractionation during NO$_3^-$ formation, and in turn to correct the raw $\delta^{15}$N-NO$_3^-$ values of cloudwater samples.

    While the N isotopic source signatures of NO$_x$ are relatively well constrained (Table B1 in

Appendix B), this is not the case for NH$_3$. We recently established a pool of isotopic source signatures of NH$_3$ in eastern China, in which livestock breeding and fertilizer application were identified to produce NH$_3$ with a $\delta^{15}$N of -29.1 ± 1.7‰ and -50.0 ± 1.8‰, respectively (Chang et al., 2016). Although fossil-fuel combustion, urban waste, and natural soils also represent potential sources of NH$_3$, their impacts are probably minor compared to that of agricultural and biomass

burning emissions, at least on a regional (or greater) scale (Kang et al., 2016). For the N isotope signature of biomass burning-derived NH$_3$ we assumed 12‰ (Kawashima and Kurahashi, 2011), a value that has also been applied in other recent isotope-based source apportionment studies (e.g., Chellman et al., 2016; Wang et al., 2017a).

## 3 Results and Discussion

**3.1 Chemical characterization of biomass-burning-influenced clouds**

    The moderate-resolution imaging spectroradiometer (MODIS) wildfire map (Fig. 1) shows that there were intensive biomass burning events occurring over mainland China, end of July 2015,

just before the study period. Moreover, analysis of the back trajectories of air masses at the study site revealed the strong influence by atmospheric transport from regions that also experienced

intensive biomass burning events shortly before the sampling campaign. It can thus be assumed that large amounts of BB-related pollutants were transported from the southwest to the sampling site at Mt. Tai. Table 1 compiles sample information and results from the chemical and isotopic analysis of cloudwater samples in this study. The concentrations of $NO_3^-$ and $NH_4^+$ ranged from 4.9 to 19.9 mg $L^{-1}$ (10.1 mg $L^{-1}$ on average), and from 4.9 to 18.0 mg $L^{-1}$ (9.1 mg $L^{-1}$ on average),

respectively, much higher than during non-BB seasons (Chen et al., 2017; Desyaterik et al., 2013; Li et al., 2017, 2018; Lin et al., 2017). Similarly, levoglucosan in our cloudwater samples varied between 12.1 and 35.1 μg $L^{-1}$ (19.9 μg $L^{-1}$ on average), and concentrations were thus one order of magnitude higher than those documented during non-BB seasons (Boone et al., 2015; Fomba et al., 2015). Although levoglucosan can be oxidized by OH radicals in the tropospheric aqueous

phase (Sang et al., 2016), it is nevertheless a reliable marker compound for BB due to its high emission factors and relatively high concentrations in the ambient aerosols (Hoffmann et al., 2010). In our study, the concentrations of $NO_3^-$ ($r^2 = 0.55$) and $NH_4^+$ ($r^2 = 0.66$) are strongly correlated with that of levoglucosan, suggesting that the pronounced increase of $NO_3^-$ and $NH_4^+$ levels observed here can at least be partly attributed to BB activities during the study period. Globally,

BB accounts for around 10% of $NH_3$ and $NO_x$ emissions (Benkovitz et al., 1996; Bouwman et al., 1997; Olivier et al., 1998; Schlesinger and Hartley, 1992).


**Table 1.** Sampling details and results of chemical and isotopic analysis for collected cloudwater samples.

| Date | Local time | $NO_3^-$ (mg $L^{-1}$) | $NH_4^+$ (mg $L^{-1}$) | levoglucosan (µg $L^{-1}$) | $\delta^{15}N$-$NO_3^-$ (‰) | $\delta^{18}O$-$NO_3^-$ (‰) | $\delta^{15}N$-$NH_4^+$ (‰) |
|------|-----------|---------|---------|-------------|------------|------------|------------|
| Aug 1 | 19:12-22:58 | 19.86 | 17.99 | 35.06 | -0.22 | 65.46 | 3.62 |
|       | 23:45-08:25 | 4.88 | 4.92 | 12.11 | -1.28 | 55.01 | 3.85 |
| Aug 2 | 09:10-12:19 | 5.57 | 5.81 | 23.60 | -0.40 | 59.84 | 0.05 |
|       | 12:55-18:30 | 13.24 | 11.51 | 18.67 | -4.18 | 55.66 | 11.34 |
|       | 20:02-23:06 | 10.06 | 7.74 | 13.63 | -3.11 | 56.63 | 12.99 |
|       | 23:48-06:12 | 6.71 | 6.59 | 16.42 | -4.92 | 54.19 | 7.33 |

## 3.2 Isotopic characterization of biomass-burning-influenced clouds

The N (and O) isotopic composition of cloudwater nitrogenous species was more ($NH_4^+$) or less ($NO_3^-$) varied (Table 1), with the average $\delta^{15}N$ values of 6.53‰ and -2.35‰ for $NH_4^+$ and $NO_3^-$, respectively. The average $\delta^{18}O$-$NO_3^-$ value was 57.80‰. These values are generally different from gas, rainwater, and aerosol values measured worldwide (Fig. 2). Various atmospheric processes can influence the isotopic composition of atmospheric nitrogenous species including:

the original emission source of $NO_x$, seasonality of oxidation pathways, isotope fractionation during transport, partitioning between wet and dry components, and spatial gradients in atmospheric chemistry (Elliott et al., 2007; Hastings et al., 2003). These aspects may affect the $\delta^{15}N$ and $\delta^{18}O$ values differentially. For example, the $\delta^{15}N$ of atmospheric $NO_3^-$ retains spatial changes in the original $NO_x$ signature quite well, in contrast to the $\delta^{18}O$. On the other hand, the

$\delta^{18}O$ most strongly depends on the oxidation chemistry and formation pathway in the atmosphere (see Equations 1-4).

At present, there are no other reports on the isotope ratios of both $NO_3^-$ and $NH_4^+$ in cloudwater, and a comparison is possible only with isotope data from precipitation and aerosol N. Recently, Vega et al. (2019) reported the $\delta^{15}N$ (-8‰ $\pm$ 2‰) and $\delta^{18}O$ (71‰ $\pm$ 3‰) values of $NO_3^-$

in fog water at a forest site in Sweden. The relatively high $\delta^{15}N$ values in our study (-2.35‰) suggest more $NO_x$ that emitted from combustion processes. In contrast, the much higher $\delta^{18}O$ values in Vega et al. (2019) indicate a much greater contribution from $O_3$ in sub-Arctic environments. In Fig. 2a, N isotopic differences for $NO_x$ sources are greater (35‰), than for $\delta^{18}O$-$NO_x$. In fact, the oxygen isotope signature of $NO_x$ is mainly chemistry-driven rather than

determined by the source (see discussion below), and thus, $\delta^{18}O$ measurements cannot be used to address the uncertainty of the $NO_x$ sources that may remain when just looking at $\delta^{15}N$ values alone. As shown in Fig. 2b, the $\delta^{15}N$ values of aerosol $NH_4^+$ are systematically higher than that of $NH_3$. Significant $\varepsilon_N$ during the conversion of gas to aerosol (up to 33‰) has been proposed to alter the $\delta^{15}N$ values during the transformation of the source ($NH_3$) to the sink (particulate $NH_4^+$). Indeed,

our compilation of previous results (Fig. 2b) reveals that particulate $NH_4^+$ (particularly in the coarse aerosol fraction) is more enriched in $^{15}N$ than $NH_3$ (by > 23‰ on average), as well as $NH_4^+$ in precipitation (by 18‰ on average). This can most likely be attributed to the preferential absorption $^{14}N$-$NH_3$ associated with washout processes during precipitation (Zheng et al., 2018). We are aware of the fact that our sample/data set used here is limited, resulting in a relatively large

uncertainty with regards to the N isotope-based source apportionment. However, all $\delta^{15}N$-$NH_4^+$ values in cloudwater samples fall within the observed range of $\delta^{15}N$-$NH_4^+$ values for fine particles (PM$_{2.5}$), providing putative evidence that $NH_4^+$ in cloudwater is primarily derived from particulate $NH_4^+$ rather than $NH_3$ absorption.

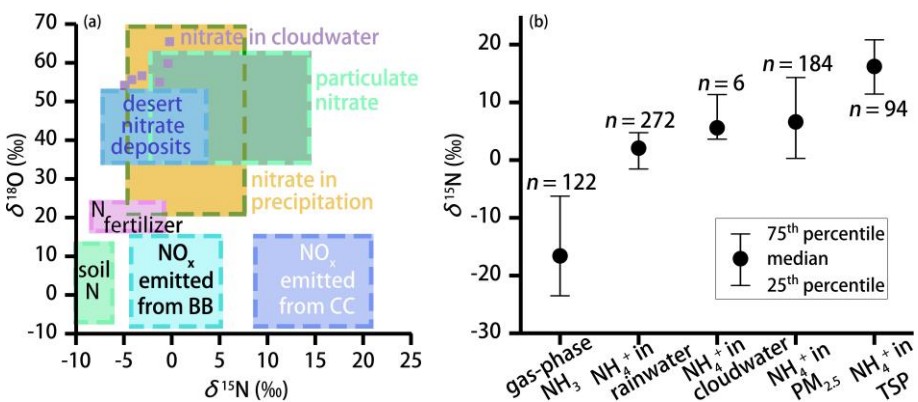

Figure 2. (a) Observed range of typical $\delta^{18}O$ and $\delta^{15}N$ values of NO$_3^-$ and NO$_x$ for different sources (adapted from Fenech et al. (2012)). BB and CC represent biomass burning and coal combustion, respectively. The red squares represent nitrate isotope data in cloudwater (this study). (b) The 25[th] percentiles, median and 75[th] percentiles for the $\delta^{15}N$ values of the ambient NH$_3$ (Chang et al., 2016; Felix et al., 2013; Savard et al., 2018; Smirnoff et al., 2012) and NH$_4^+$ in precipitation (Fang et al., 2011; Leng et al., 2018; Yang et al., 2014; Zhang et al., 2008), cloudwater (this study), PM$_{2.5}$ (particulate matter with aerodynamic diameter less than 2.5 μm; (Lin et al., 2016; Park et al., 2018; Proemse et al., 2012; Smirnoff et al., 2012), and TSP (particulate matter with aerodynamic diameter less than 100 μm (Kundu et al., 2010; Savard et al., 2018; Yeatman et al., 2001) are shown.

### 3.3 Isotope-based assessment of the sources and formation of nitrogenous species in clouds

Using the MixSIR model, the relative contribution of four NH$_3$ sources to NH$_4^+$ can be calculated, based on the isotope data of ambient $\delta^{15}N$-NH$_4^+$, and considering the N fractionation and prior information on the site. As upper limit for the N isotope enrichment factor associated with the conversion of NH$_3$ to NH$_4^+$ ($\varepsilon_{NH_4^+-NH_3}$), we assumed 33‰ when using MixSIR, but also considered lower values for $\varepsilon_{NH_4^+-NH_3}$ (Fig. 3a) (given the conflicting evidence with regards to

$\varepsilon_{NH_4^+-NH_3}$ ; e.g., Deng et al., 2018; Li et al., 2012). Dependent of the choice for $\varepsilon_{NH_4^+-NH_3}$ (between

0‰ to 33‰ proposed by Heaton et al. (1997)) the relative contribution of biomass burning, fertilizer application, and livestock breeding to $NH_4^+$ in cloudwater ranges from 25.9% to 85.4%, 5.9% to 37.0%, and 8.7% to 85.4%, respectively. Irrespective of the uncertainty related to $\varepsilon_{NH_4^+-NH_3}$,

the measurement of levoglucosan provides compelling evidence that biomass burning represents an important $NH_3$ source, independently validating our isotope approach. Our sampling site was located in the North China Plain, also known as the granary of China. Although non-agricultural $NH_3$ emissions like on-road traffic are important in the urban atmosphere (Chang et al., 2016), their contribution must be considered insignificant with respect to fertilizer application and

livestock breeding in this region (Kang et al., 2016). Besides, coal-based heating in China is suspended during summertime, and coal combustion has been demonstrated to be a minor contributor of total $NH_3$ emissions (Li et al., 2016a). Hence the partitioning between the three main $NH_3$ sources appears plausible. Moreover, existing emission inventory data confirm that the ratio of $NH_3$ emissions in North China Plain from livestock breeding (1658 kt) and fertilizer application

(1413 kt) was 1.17 (Zhang et al., 2010), which is very close to our estimate (between 0.98 and 1.14) when $\varepsilon_{NH_4^+-NH_3} \geq 25$‰. For a $\varepsilon_{NH_4^+-NH_3}$ range that we consider most plausible (i.e. between 25‰ and 33‰), the relative cloudwater $NH_4^+$ source partitioning between biomass burning, fertilizer application, and livestock is 32.9 ± 4.6%, 32.9 ± 3.0%, and 34.2 ± 1.6%, respectively (indicated as red square in Fig. 3a). It is important to note that there was large uncertainty in the

evaluation of biomass burning, which was partly ascribed to the lack of localized isotopic source signatures in China. In addition, the isotopic fractionation from the conversion of $NH_3$ to $NH_4^+$ was simplified in this study, and it was not possible to incorporate all of the possible equilibrium and kinetic fractionation scenarios.

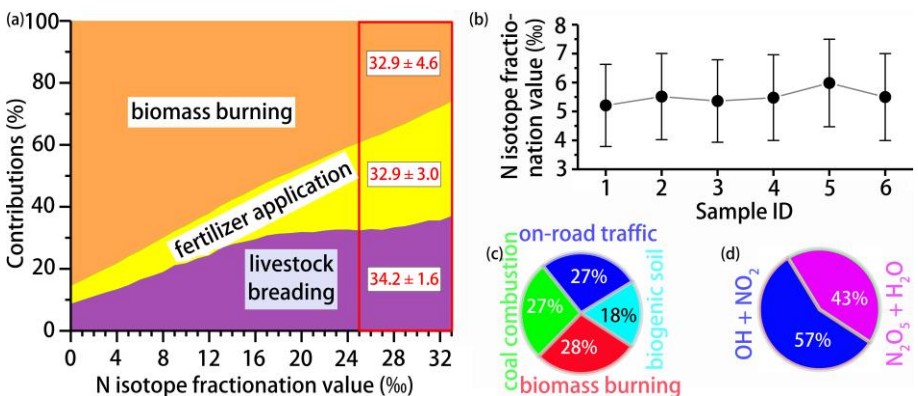

**Figure 3**. (**a**) Source partitioning estimates for $NH_4^+$ in cloudwater as a function of $\varepsilon_{NH_3 \rightarrow NH_4^+}$. The red square highlights the best-guess estimates based on $\varepsilon_{NH_4^+ - NH_3} \geq 25‰$). (**b**) Whisker plot of the N fractionation for the conversation of $NO_x$ to $NO_3^-$ ($\varepsilon_{NO_3^- - NO_x}$) calculated by the computational quantum chemistry (CQC) module. The upper line, dot, and bottom line indicate the 25th percentile, median, and 75th percentile, respectively. Refer to Table 1 for sample ID. (**c**) Overall contribution of various $NO_x$ sources to $NO_3^-$ in cloudwater as estimated by the MixSIR model. (**d**) Overall contribution of the two dominant pathways to $NO_3^-$ formation in cloudwater, as estimated by the MixSIR model.

The computational quantum chemistry (CQC) module in MixSIR has been proven a robust tool to quantify the N isotope enrichment factor during $NO_x$-$NO_3^-$ conversion ($\varepsilon_{NO_3^- - NO_x}$) (e.g., Zong et al., 2017, Chang et al., 2018). Cloudwater sample-based data from this study reveal that $\varepsilon_{NO_3^- - NO_x}$ values fall into a small range (5.21‰ to 5.98‰) (Fig. 3b), suggesting robust isotope effects during the N isotopic exchange reactions. Knowing $\varepsilon_{NO_3^- - NO_x}$, the overall contribution of various $NO_x$ sources to $NO_3^-$ in cloudwater can be estimated (Fig. 3c). As was expected, biomass burning was the largest contributor (28.2 ± 2.7%), followed by on-road traffic (27.1 ± 2.2%), coal combustion (26.8 ± 3.4%), and biogenic soil (17.9 ± 3.9%). The fundamental importance of

biomass burning-emitted $NO_x$ to $NO_3^-$ in cloudwater is supported by the observed correlation between the concentrations of levoglucosan and biomass burning-derived $NO_3^-$ ($r^2 = 0.66$). The average contribution ratio of coal combustion and on-road transportation to $NO_x$ emissions in our study (0.99) is slightly lower than that calculated from regional emission inventories (9.0 Tg/7.4

355    Tg = 1.22) (Zhao et al., 2013). The apparent difference is likely real, and reflects the fact that $NO_x$ emissions by anthropogenic activities changed significantly since 2010: a 17% total emission decrease between 2010 and 2017 can primarily be attributed to upgraded emission standards and new "ultra-low emission" techniques in the coal-fired power plant sector, given that traffic-emitted $NO_x$ likely increased as a consequence of the continuous expansion of auto trade market during

the last decade (Chang et al., 2018). In turn, our source partitioning estimate probably reflects the most updated status of $NO_x$ emissions in China, where transportation-related $NO_x$ emissions have reached levels that are comparable to $NO_x$ emissions by coal combustion. In this regard, our study demonstrates that Bayesian-based isotopic mixing modeling can be an effective and timely approach to track rapid emissions changes of $NO_x$ in a fast-developing country like China.

Using the measured $\delta^{18}O$ and equations 1-4 (and the assumptions above), we can calculate $\gamma$, and the relative importance of the two oxidation pathways of $NO_3^-$ formation (Fig. 3d). On average, 57% $NO_3^-$ formation can be attributed to the "$NO_2 + OH$" pathway, and 43% to the "$N_2O_5 + H_2O$" pathway. In the low-latitude regions, where atmospheric OH concentrations are highest, particulate $NO_3^-$ production via the "$NO_2 + OH$" pathway predominates (up to 87%) (Alexander

et al., 2009). Sampling during summertime, oxidation of $NO_2$ through OH was expected to be the dominant pathway of nitrate formation, in accordance with observations from the subtropics (Hastings et al., 2003). However, our results highlight that $N_2O_5$ hydrolysis can be an almost

equally important process as the oxidization of $NO_2$ with OH with regards to the $NO_3^-$ formation in cloudwater (Wang et al., 2017b).

**4 Conclusions**

In this study, we measured the isotopic composition of nitrogenous species in cloudwater at the summit of Mt. Tai during a long-lasting biomass burning event, in order to investigate the sources and processes involved in cloudwater $NO_3^-$/$NH_4^+$ formation, and in turn to test our isotope-balance approach to constrain N source partitioning in cloudwater. Using a Bayesian isotope mixing model, the $\delta^{15}N$-based estimates confirm that at least transiently biomass burning related $NH_3$ and $NO_x$ emissions is a major source of cloudwater N. Moreover, our data are in accordance with regional emission inventories for both $NH_3$ and $NO_x$, validating the Bayesian isotope mixing model approach. Based on cloud water nitrate $\delta^{18}O$ measurements, the reaction of $NO_2$ with OH turned out to be the dominant pathway to form cloud nitrate, yet the contribution from the heterogeneous hydrolysis of $N_2O_5$ to $NO_3^-$ is almost equally important. Our study underscores the value of cloud-water dissolved inorganic nitrogen isotopes as carrier of quantitative information on regional $NO_x$ emissions. It sheds light on the origin and production pathways of nitrogenous species in clouds and emphasizes the importance of BB-derived nitrogenous species as cloud condensation nuclei in China's troposphere. Moreover, it highlights the rapid evolution of $NO_x$ emissions in China. Despite an overall reduction in total anthropogenic $NO_x$ emission due to effective emission control actions and stricter emission standards for vehicles, the relative contribution of transportation to total $NO_x$ emissions has increased over the last decade and may already have exceeded emissions from the power sector.

## Appendix A: Bayesian isotopic mixing model

The Bayesian mixing model makes use of stable isotope data to determine the probability distribution of source contributions to a mixture, explicitly accounting for uncertainties associated with multiple sources, their isotopic signatures, and isotope fractionation during transformations. The model has been widely used in ecological studies, such as food-web analyses. In Bayesian theorem, the contribution of each source is calculated based on mixed data and prior information,

such that:

$$P ( f_q | \text{data} ) = \theta(\text{data} | f_q) \times p ( f_q ) / \sum \theta(\text{data} | f_q) \times p ( f_q )$$

where $\theta(\text{data}|f_q)$ and $p(f_q)$ refer to the likelihood of the given mixed isotope signature, and the pre-determined probability of the given state of nature, based on prior information, respectively. The denominator represents the numerical approximation of the marginal probability of the data. In a

405 Bayesian model (stable isotope in R; SIAR), isotope signatures from the mixed data pool are assumed to be normally distributed. Uncertainty in the distribution of isotope sources and associated isotope fractionation during transformations are factored into the model by defining respective mean ($\mu$) and standard deviation ($\sigma$) parameters. Prior knowledge about proportional source contributions ($f_q$) is parametrized using the Dirichlet distribution, with an interval of [0, 1].

To assess the likelihood of the given $f_q$, the proposed proportional contribution is combined with a user-specified isotope distribution of sources and their associated isotope effects to develop a proposed isotope distribution for the mixture. The probability of fractional source contributions ($f_q$) is calculated by the Hilborn sampling-importance-resampling method.

## Appendix B: Isotopic signatures of NOx emitted from various sources

**Table B1.** Typical $\delta^{15}$N-NO$_x$ values for coal combustion, transportation and biomass burning, and soils based on literature values.

| Source types | Mean (‰) | Standard(‰) | Number | Reference |
|---|---|---|---|---|
| Coal combustion | 13.72 | 4.57 | 47 | Felix et al., 2012, 2015 |
| Transportation | -7.25 | 7.80 | 151 | Walter et al., 2015a, b; Heaton et al., 1997 |
| Biomass burning | 1.04 | 4.13 | 24 | Fibiger and Hastings, 2016; Felix and Elliott, 2013 |
| Biogenic soil | -33.77 | 12.16 | 6 | Hastings et al., 2009; Felix et al., 2012 |

## Data availability

All data used to support the conclusion are presented in this paper. Additional data are available upon request. Please contact the corresponding authors (Yanlin Zhang (dryanlinzhang@outlook.com) and Jianmin Chen (jmchen@fudan.edu.cn)).

## Acknowledgments

This study was supported by the National Key R&D Program of China (Grant no. 2017YFC0212700, task 1 and 4), National Natural Science Foundation of China (Grant nos. 41975166, 41705100, 91644103), the Provincial Natural Science Foundation of Jiangsu (Grant nos. BK20180040, BK20170946), University Science Research Project of Jiangsu Province (17KJB170011), the University of Basel research funds, the Priority Academic Program Development of Jiangsu Higher Education Institutions (PAPD), the Program for Changjiang Scholars and Innovative Research Team in University of Ministry of Education of China (PCSIRT), and the Opening Project of Shanghai Key Laboratory of Atmospheric Particle Pollution and Prevention (LAP[3]).

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
