# Peer review of "Isotopic Constraints on the Atmospheric Sources and Formation of Nitrogenous Species in Biomass-Burning-Influenced Clouds"

_Atmospheric Chemistry and Physics, 2018_

## Short Comment (SC1) · 3 Feb 2019

Comments for Chang et al. "Isotopic Constraints on the Atmospheric Sources and Formation of Nitrogenous Species in Biomass-Burning-Influenced Clouds"

In this study, the authors analyzed stable nitrogen and oxygen isotope ratios timeseries of a biomass burning event at a mountaintop site in eastern China to investigate the sources and formation processes of nitrogenous species in cloudwater. A theoretical approach to calculate N isotope effects is used. While extensive studies of nitrogen-

containing aerosol isotopic compositions have been conducted worldwide, this study represents the first attempt to isotopically explore what happens to nitrogenous species in cloudwater. My major concern is relatively short discussion regarding their data and results. I will be glad to recommend a final publication after the following questions can be addressed. 1. I am not questioning the novelty of this work, but the authors should put more efforts to clearly describe the novelty of their study in the abstract or introduction section. 2. Since the isotope data reported in the MS are the first of its kind, thus the authors should discuss more about the data in section 3.2. 3. The authors assess the contributions of potential sources and processes with exact numbers. I suggest the authors to discuss more about the uncertainties of these results.

I have a few specific comments: 1. Line 52-56: the authors should double-check the references used here. 2. Line 127-129: brief introduction about the chemical analysis is need. 3. Line 205: "burring" should be changed as "burning". 4. Please note that ÔŚ represents isotope enrichment factor. Keep a consistent expression in the text.

---

## Referee Comment (RC1) · Anonymous Referee #1 · 12 Mar 2019

Chang et al. reported the isotopic composition of nitrogen species in cloud water at a mountain site in North China, during a biomass burning event. They analyzed the data with isotope mixing model and CQC module to investigate the sources and formation mechanisms of nitrogenous species in cloud water. The study contributes to the growing body of isotope measurements around the world, and the methods could be useful for source and chemical process analysis. The manuscript can be improved by adding more in-depth analysis/discussion on the data. Also, there are a number of places which need to be modified or clarified. The figures are not in good quality, and

most of the figures are too fuzzy to see clearly.

It would be interesting also to examine the isotope information in the aerosols collected at the same site during the same period. The comparison between cloud and aerosol would provide more useful and meaningful information to further understand the sources and formation mechanisms of these nitrogenous species.

I also agree with the comments of Chen, 2019, and the author should provide more information and assessment on the uncertainties of the results. The exact numbers may not be representative given the small number of samples.

In addition, I would suggest the author to further examine the influences of cloud water content and cloud process on the change of isotopic data in the same cloud event. The changes and variations of these $\delta$ values may not necessarily result from the differences in source or formation mechanisms. In section 3.3, all the equations and descriptions (page 15-17) are the same as in Chang et al., 2018. I would suggest the author condense this part and include it in the methodology section. More details may be provided in SI.

Specific comments: Line 111. Were the six cloud water samples collected during two or three isolated cloud events?

Line 129. The treatment and analytical protocol can be the same as previous literature, the detection limits and errors, reproducibility, recovery rates would be varied for different research. Please clarify.

Line 141. Please clarify how the sample values were corrected.

Line 189 and Table 1. Better to use consistent units.

Line 193. It is better to compare with the results at Mt Tai during non-BB seasons. The comparison with other regions cannot be used to support the author's statement.

Section 3.2. As mentioned above, did the variations reflect different periods of the

cloud event? Will the cloud evolution process affect the results/conclusion of source analysis?

Line 209-214. This information can be included in the introduction, but it seems not useful to explain the measurement data in this study.

Line 223-226. The statement is too speculative and does not scientifically sound. Without the isotope analysis, you can still conclude that there is a link between different process.

Line 232. Please provide reference or evidence to support the statement.

Line 233-235. Was this 'no significant difference' found in this measurement? Otherwise, how could the author deduce the conclusion here?

Section 3.3. More information on the uncertainty of the numbers is needed. In addition, considering the uncertainty, the significance digit can be rounded.

Line 289-291. The measurement data in six samples can be affected by different factors, and doesn't necessarily to be the same as the emission inventory.

Line 291-296. These descriptions are generally correct and reasonable. However, this discussion seems not directly link to the data/result in this study.

Line 326-328. It is better to compare with low-altitude data in the same region. Otherwise, the comparison may not make sense.

Line 345-346. It seems the author did not discuss the production pathways in clouds, which can be very interesting if the author can do further analysis in this area.

---

## Referee Comment (RC2) · Anonymous Referee #2 · 24 Mar 2019

This study utilizes the isotopic composition of nitrate and ammonium captured in cloud-water during a major biomass burning event to delineate the sources and chemistry that contribute to the inorganic N concentrations. Overall, the methods are sound and the study yields interesting results. The interpretation is well constructed, but there are a number of aspects that need to be better justified and better referenced to make the study's findings stronger. Please see below.

As motivation for the work, the authors invoke the difficulty of addressing tropospheric cloud formation and its importance for radiative forcing of climate. Further, they state

the importance of cloud properties linked to aerosol and precipitation chemistry. In the end, however, there is not a direct connection back to how understanding and quantifying the sources of nitrate and ammonium in this case actually link to clouds, cloudwater or their properties. It would be useful if the authors could make a stronger link in the end of the work, or edit the motivation to better fit the outcomes of the study.

There are several instances of discussing the link between nitrate formation pathways and the oxygen isotopic composition of nitrate without references. This is fairly well established in the literature already – with an emphasis on D17O in works such as Michalski et al. (2003, 2004, 2005, 2011 etc), Alexander et al. (2009), Morin et al. (2011), and an emphasis on d18O in works such as Hastings et al. (2003, 2004, 2013), Elliott et al. (2009). Many of these works are already cited by the authors, but it should be made more clear in several places in the manuscript where the framework for interpreting d18O comes from. The text reads as if this current study is establishing that the oxygen isotopic signature (d18O) comes from chemical formation of nitrate – yet this was established as early as in Hastings et al. (2003). For instance, please add references to the first line on page 5; to line 216 on page 11; discuss more how equations 1-4 are generated and the fact that they are largely based upon previous work (for instance Hastings et al. 2003 and Michalski et al. 2011!).

Related to this, the authors should compare the quantification of the oxidation chemistry with the expectations set forth in Alexander et al., 2009. While Alexander et al. is a modeling study, with significant uncertainties, it still would be helpful to understand a greater context for the findings in this study and whether they reflect what we expect in terms of atmospheric chemistry in the extratropics or whether the observations are suggesting something new that modelers need to be thinking about. Furthermore, it needs to be justified why only the OH and N2O5 pathways for HNO3 formation are used (page 15). In biomass burning plumes (and in initial emissions measured in the laboratory), a great deal of organics are also found. This makes it likely that RO2 and/or H-abstraction pathways would be important as well. How can this not be included here?

Also, what is assumed for the initial d18O-NOx in equation 2. It appears there are more unknowns than knowns for equation 2 as currently discussed in the manuscript. In the end, it is hard to justify the conclusion (like 330) that N2O5 is particularly important in cloudwater without further justification as to why only OH and N2O5 pathways are important to begin with. (On BB emissions and plumes: this is just a sampling of relevant literature and there may be better studies to reference that are more relevant to the field location in China in this study – Akagi et al., 2013; Alvarado et al. 2009a and 2009b and 2015; Burling et al., 2011; Yokelson et al. 2008 and references therein)

What is the expectation for lifetime of the nitrate and ammonium in this study? Models suggest that the lifetime of NOx should be $\sim 1-1.5$days at this latitude (Levy et al., 1999) and that the lifetime of nitrate and ammonium aerosols (globally) is on the order of 3-5 days (Xu and Penner, 2012). The other tracers (e.g. levoglucosan) and transport patterns help to establish the likely influence of the biomass burning smoke, but this should be better integrated in the manscuript with expectation for the formation and transport of nitrate and ammonium, since the different tracers would be expected to have different lifetimes.

The methods section should include the details of the HYSPLIT back-trajectories – what meteorological dataset is used? At what heights? Etc. Also, it should be justified why 48-hours is used and tied to the question above regarding expected lifetimes for nitrate and ammonium.

On page 8, line 165-167 it is stated that the raw d15N-NO3- data was corrected to the calculate the N isotope fractionation. How is this done? What values were found? How does this compare with expectations in the literature for the computed fractionations (i.e. Walters and Michalski (2015), Walters et al. (2016) and Chang et al. (2018))?

There is discussion of the fractionation of conversion from NH3 gas to NH4+ aerosol on pages 11-13. The authors should also consider newer work on this subject – Walters et al. (2019) as it is highly relevant to the discussion here.

Additional specific comments:

Line 26: This phrasing is awkward – perhaps change to "However, this challenge is difficult to address quantitatively based on the sole use of bulk chemical properties."

Line 94: there is one study in Hawaii that includes fog deposition at a high altitude site that might be relevant here (Carillo et al., 2002). Even if the data is not directly comparable it should be considered that the current manuscript is not the only measurements that exist.

Line 225-230: The discussion of the 33 per mil isotope effect found by Heaton should be cited as such. It is mentioned here as if this is well established, but Heaton himself in this work considers the study preliminary. Also please see Walters et al. (2019).

Line 256: The idea that this validates the approach seems circular in logic. The isotope ranges and fractionations incorporated into the mixing model begins with assumptions about what sources should be important; the fact that the modeling then yields the conclusion that biomass burning is an important source follows from the initial assumptions, it does not in fact justify those initial assumptions. Additional data of other kinds that suggest the same conclusion are more appropriate for making this claim.

Supplement: please consider more recent observations of d15N-NOx for vehicles and especially for soils (Miller et al. 2017, 2018; Yu and Elliott, 2017) and the issues related to previous collection techniques for source signatures (Fibiger et al., 2014). The references listed for the biogenic soil emissions are not really relevant.

References cited:

Akagi, S. K., Yokelson, R. J., Burling, I. R., Meinardi, S., Simpson, I., Blake, D. R., McMeeking, G. R., Sullivan, A., Lee, T., Kreidenweis, S., Urbanski, S., Reardon, J., Griffith, D. W. T., Johnson, T. J. and Weise, D. R.: Measurements of reactive trace gases and variable O3 formation rates in some South Carolina biomass burning plumes, Atmos Chem Phys, 13(3), 1141–1165, doi:10.5194/acp-13-1141-2013, 2013.

[Figure]

Alvarado, M. J. and Prinn, R. G.: Formation of ozone and growth of aerosols in young smoke plumes from biomass burning: 1. Lagrangian parcel studies, J. Geophys. Res. Atmospheres, 114(D9), doi:10.1029/2008JD011144, 2009.

Alvarado, M. J., Wang, C. and Prinn, R. G.: Formation of ozone and growth of aerosols in young smoke plumes from biomass burning: 2. Three-dimensional Eulerian studies, J. Geophys. Res. Atmospheres, 114(D9), doi:10.1029/2008JD011186, 2009.

Alvarado, M. J., Lonsdale, C. R., Yokelson, R. J., Akagi, S. K., Coe, H., Craven, J. S., Fischer, E. V., McMeeking, G. R., Seinfeld, J. H., Soni, T., Taylor, J. W., Weise, D. R. and Wold, C. E.: Investigating the links between ozone and organic aerosol chemistry in a biomass burning plume from a prescribed fire in California chaparral, Atmos Chem Phys, 15(12), 6667–6688, doi:10.5194/acp-15-6667-2015, 2015.

Burling, I. R., Yokelson, R. J., Akagi, S. K., Urbanski, S. P., Wold, C. E., Griffith, D. W. T., Johnson, T. J., Reardon, J. and Weise, D. R.: Airborne and ground-based measurements of the trace gases and particles emitted by prescribed fires in the United States, Atmos Chem Phys, 11(23), 12197–12216, doi:10.5194/acp-11-12197-2011, 2011.

Carrillo, J. H., M. G. Hastings, D. M. Sigman, and B. J. Huebert, Atmospheric deposition of inorganic and organic nitrogen and base cations in Hawaii, Global Biogeochem. Cycles, 16(4), 1076, doi:10.1029/2002GB001892, 2002.

Fibiger, D.L., M.G. Hastings, A.F. Lew, R.E. Peltier, Collection of NO and NO2 for isotopic analysis of NOx emissions, Analytical Chemistry, 86 (24), 12115–12121, doi: 10.1021/ac502968e, 2014.

Michalski, G. S.K. Bhattacharya, and D. F. Mase, Oxygen Isotope Dynamics of Atmospheric Nitrate and Its Precursor Molecules (Chapter 30), in M. Baskaran (ed.), Handbook of Environmental Isotope Geochemistry, Advances in Isotope Geochemistry, DOI 10.1007/978-3-642-10637-8_30, # Springer-Verlag Berlin Heidelberg 2011.

Miller, D.J., J. Chai, F. Guo, C.J. Dell, H. Karsten, and M.G. Hastings, Isotopic composition of in situ soil NOx emissions in manure fertilized cropland, Geophysical Research Letters, 45(21), 12058-12066, https://doi.org/10.1029/2018GL079619, 2018.

Miller, D.J, P.K. Wojtal, S.C. Clark, and M.G. Hastings, Vehicle NOx emission plume isotopic signatures: Spatial variability across the eastern United States, J. Geophys. Res. Atmos., 122, doi:10.1002/2016JD025877, 2017.

Levy, H., II, W. J. Moxim, A. A. Klonecki, and P. S. Kasibhatla, Simulated tropospheric NOx: Its evaluation, global distribution and individual source contributions, J. Geophys. Res., 104, 26,279 – 26,306, 1999. – see the appendix figure

Walters, W.W, J. Chai, M. G. Hastings, Theoretical Phase Resolved Ammonia-Ammonium Nitrogen Equilibrium Isotope Exchange Fractionations: Applications for Tracking Atmospheric Ammonia Gas-to-Particle Conversion, ACS Earth and Space Chemistry, 3, 79-89, DOI: 10.1021/acsearthspacechem.8b00140, 2019.

Xu, L. and J.E. Penner, Global simulations of nitrate and ammonium aerosols and their radiative effects, Atmos. Chem. Phys., 12, 9479–9504, www.atmos-chem-phys.net/12/9479/2012/, 2012. – see Tables 4 and 5

Yokelson, R. J., Christian, T. J., Karl, T. G. and Guenther, A.: The tropical forest and fire emissions experiment: laboratory fire measurements and synthesis of campaign data, Atmos Chem Phys, 8(13), 3509–3527, doi:10.5194/acp-8-3509-2008, 2008.

Yu, Z. and E. Elliott, Novel Method for Nitrogen Isotopic Analysis of Soil-Emitted Nitric Oxide, Environ. Sci. Technol., 51, 6268−6278, DOI: 10.1021/acs.est.7b00592, 2017.

---

## Referee Comment (RC3) · Anonymous Referee #3 · 23 Apr 2019

MS No.: acp-2018-1196 Title: Isotopic Constraints on the Atmospheric Sources and Formation of Nitrogenous Species in Biomass-Burning-Influenced Clouds

The objective of the submitted manuscript "Isotopic Constraints on the Atmospheric Sources and Formation of Nitrogenous Species in Biomass-Burning-Influenced Clouds" was to apply stable isotope techniques to determine sources and pathways of inorganic nitrogen in cloudwater. Although the presented work provides a very limited data set, it is the first of its kind to measure $\delta$15N-NH4+ in cloudwater and second of its kind to measure $\delta$15N-NO3- in cloudwater and apply these values to determine poten-

tial sources of the nitrogen species. Understanding the dynamics of nitrogen species in cloudwater is important since cloudwater has recently been reported to be a significant contributor to nitrogen deposition in various regions. If the authors adequately address the issues outlined below I believe the work can be a valuable addition to the current atmospheric nitrogen literature and should be accepted to Atmospheric Chemistry and Physics.

Comments:

Line 30: The authors state "...measured for the first time the isotopic compositions of cloudwater nitrogen species...". This may have been the case during the measurements or manuscript submission process but there has been a recent paper published that would be considered a cloudwater study of nitrate isotopes (Vega et al., 2019). However, it is likely the first with ammonium isotopes and nitrate isotopes in this region. The instances alluding to the "first time" or novelty of the data should be changed accordingly.

The % deviation associated with the authors' source apportionment model will significantly vary depending on the range in nitrogen emission sources. The authors use $\delta$15N-NH3 signatures of -29.1 $\pm$ 1.7‰ and -50.0 $\pm$ 1.8‰ for livestock and fertilizer emission sources. According to the literature this source range and standard deviation isn't realistic and likely doesn't reflect source ranges that occur due to various chemical and physical factors associated creating this source signature. Elliott et al. 2019 has a thorough compilation of literature $\delta$15N-NH3 signatures. The authors did an adequate job when compiling the $\delta$15N-NOx source signatures and the mixing model for NH3 would benefit from a similar approach.

The authors dismiss fossil fuel combustion (vehicles and power plants) as an emission source in this study region when discussing contribution to atmospheric NH3 but argue for its significance in this region when discussing NOx source apportionment. For NH3: "Although fossil-fuel combustion, urban waste, and natural soils also represent potential sources of NH3, their impacts are probably minor compared to that of agricultural and biomass burning emissions, at least on a regional (or greater) scale (Kang et al., 2016). Although non-agricultural NH3 emissions like on-road traffic are important in the urban atmosphere (Chang et al., 2016), their contribution must be considered insignificant with respect to fertilizer application and livestock breeding in this region (Kang et al., 2016). Besides, coal based heating in China is suspended during summertime, and coal combustion has been demonstrated to be a minor contributor of total NH3 emissions (Li et al., 2016a)." For NOx: "As was expected, biomass burning was the largest contributor (28.2 ± 2.7%), followed by on-road traffic (27.1 ± 2.2%), coal combustion (26.8 ± 3.4%), and biogenic soil (17.9 ± 3.9%). "….NOx emissions by anthropogenic activities changed significantly since 2010: a 17% total emission decrease between 2010 and 2017 can primarily be attributed to upgraded emission standards and new "ultra-low emission" techniques in the coal-fired power plant sector, given that traffic-emitted NOx likely increased as a consequence of the continuous expansion of auto trade market during the last decade." The argument for and against these fossil fuel sources, as outlined above, may confuse the reader especially since high NH3 concentration have been linked to traffic and the authors contribute a significant amount of NOx to vehicles. Also, the authors mention "ultra-low emissions" techniques when referring to NOx contributions and these techniques would include SCR technology in coal combustion plants that lead to NH3 emissions. The authors should clear up their arguments in this section so there aren't contradictory statements or so the readers understands why the arguments seem contradictory.

Line 125: Additional inorganic ion concentration measurements are mentioned. Was NO2- also measured? If so, was the concentration significant compared to NO3-. It will also be measured in the isotope analysis and will contribute to the $\delta$15N-NO3- value reported. Was NO2- removed before $\delta$15N-NO3- analysis?

Line 197: State the significance of the correlation using p-value. The authors don't refer to the very strong correlation coefficient between NO3- and NH4+ although this

could help argue for the similar primary source (BB).

Line 205: change "biomass-burring" to "biomass-burning"

Line 207: Are these $\delta$15N averages concentration-weighted? The weighted average would be a better representation of the overall source contribution.

Line 223: The discussion comparing $\delta$15N- NH3/4 in gas, cloudwater, rain, and particulates may be overstated due to the small sample size. The authors should at least remind the reader that this is a small sample set and these comparisons are preliminary. Also the authors can now compare to the Vega et al., fogwater values.

Line 231: "This can most likely be attributed to the preferential absorption 14N-NH3 associated with washout during precipitation." Is there a reference to this? Is this trend observed in literature?

Line 250: The authors discuss equilibrium fractionation but do not address the kinetic fractionation that is predicted to have an opposite fractionation effect ($\varepsilon$ = 28‰ (Pan et al., 2016). The authors should make the reader aware of this pathway and discuss why they assumed it is insignificant if they are not taking it into account when investigating the $\delta$15N data.

Line 268: When taking into account the literature range and overlap of fertilizer and livestock waste emission $\delta$15N-NH3 values and the fact that both sources originate from source pools (waste and fertilizer N) with similar $\delta$15N values and are the product of similar fraction effects, is it realistic to treat these as separate sources rather than just an overall agricultural source?

Line 301 and 39: OH oxidation is mentioned as the dominate pathway. The wording here should be changed since the results do not indicate it is dominate. Also, it would be expected that OH is the dominate pathway during this sampling period, why do the authors think it wasn't in this particular case?

References:

Elliott, E.M., Yu, Z., Cole, A.S. and Coughlin, J.G., 2019. Isotopic advances in understanding reactive nitrogen deposition and atmospheric processing. Science of The Total Environment, 662, pp.393-403.

Pan, Y., Tian, S., Liu, D., Fang, Y., Zhu, X., Zhang, Q., Zheng, B., Michalski, G. and Wang, Y., 2016. Fossil fuel combustion-related emissions dominate atmospheric ammonia sources during severe haze episodes: Evidence from 15N-stable isotope in size-resolved aerosol ammonium. Environmental science & technology, 50(15), pp.8049-8056.

Vega, C.P., Mårtensson, E.M., Wideqvist, U., Kaiser, J., Zieger, P. and StrÖm, J., 2019. Composition, isotopic fingerprint and source attribution of nitrate deposition from rain and fog at a Sub-Arctic Mountain site in Central Sweden (Mt Åreskutan). Tellus B: Chemical and Physical Meteorology, pp.1-19.

---

## Author Comment (AC1) · 2 Aug 2019

Please refer to the attached pdf file for the reply letter and the MS with tracked changes.

Please also note the supplement to this comment:
https://www.atmos-chem-phys-discuss.net/acp-2018-1196/acp-2018-1196-AC1-supplement.pdf
* * *
[Figure]

2018.

---

## Author Comment (AC3) · 2 Aug 2019

We greatly thank Prof. Yingjun Chen (P25-27) and three anonymous reviewers (P1-8 for referee #1; P9-14 for referee #2; P15-24 for referee #3) for their insightful comments, which we've fully addressed (in bold) point by point in our reply letter below. The revised MS with tracked changes (P28-59) is attached after the reply letter. We are deeply sorry for the delay due to the family affairs of the first author.

**5**

**Anonymous Referee #1 General Comments:**

10

15

20

1. Chang et al. reported the isotopic composition of nitrogen species in cloud water at a mountain site in North China, during a biomass burning event. They analyzed the data with isotope mixing model and CQC module to investigate the sources and formation mechanisms of nitrogenous species in cloud water. The study contributes to the growing body of isotope measurements around the world, and the methods could be useful for source and chemical process analysis. The manuscript can be improved by adding more in-depth analysis/discussion on the data. Also, there are a number of places which need to be modified or clarified. The figures are not in good quality, and most of the figures are too fuzzy to see clearly.

Many thanks for the recognition of our work. We appreciate the constructive suggestions, which have helped to greatly improved our MS. More in-depth analysis/discussion on the data have been added in the text, and where requested text sections have been clarified in the revised MS (see details below).

We are sorry for the difficulties with the figure quality experienced by referee #1. All three original figures have been replotted in the revised MS:

Figure 1

**Original version:**

1

**Revised version:**

---

## Author Response (AR2)

Dear Prof. Nizkorodov,

We thank you again for handling our manuscript. All your comments have been fully addressed. Please see our point-by-point reply below. The revised MS with tracked changes is also attached after the reply letter.

1. Line 112-113: equations in PDF do not come out well; appear elevated. I find that notation is unnecessarily complicated in the equations on this page but if this is what people use in this community, it is fine.

**Reply: Now the elevated equations have been**

2. Line 130: End -> At the end

**Reply: Corrected accordingly.**

3. Line 174: "ppb" refers to par per billion by weight in solution? Atmospheric chemistry readers

15   may confuse it with part per billion by volume, so perhaps it is best to define.

**Reply: We agree that for the measurements of aerosol species and gaseous pollutants, our community prefers to use μg m$^{-3}$ and ppb as unit, respectively, and this is not the case for cloudwater. We've defined "ppb" as you suggested in the revised MS.**

20   4. Line 199: equation number missing

**Reply: Added in the revised MS.**

5. Line 259: variant -> varied

**Reply: Corrected accordingly.**

6. Figure 2: I would make points from your study a bit more prominent (they are hard to notice against bright colors of the rectangular regions)

**Reply: Please check the revised version below**

[Figure]

7. The supporting information file is too short to justify its existence. It only consists of one page of text, which overlaps partly with the info reported in the main text, and one Table S1. I suggest that table S1 should move to the main text and the supporting information file should be eliminated. Some people use Appendix at the end of the paper instead of supporting

35    information section; this is also a possibility to consider.

**Reply: Agree. We've moved Text S1 and Table S1 to the text as Appendix A and Appendix B, respectively.**

[revised manuscript text omitted]